# Optical Investigation of the Limits of Modeling the Nonlinear Elastic Behavior of PA6 Using Linear Elastic Material Models

**Máté File [1,\*], Tamás Mankovits [1] and Dávid Huri [1,2]**

1   Department of Mechanical Engineering, Faculty of Engineering, University of Debrecen,
    H-4028 Debrecen, Hungary; tamas.mankovits@eng.unideb.hu (T.M.); huri.david@eng.unideb.hu (D.H.)
2   Faculty of Informatics, Doctoral School of Informatics, University of Debrecen, H-4028 Debrecen, Hungary
\*   Correspondence: mate.file@eng.unideb.hu

**Featured Application: In the design of engineering polymeric products, the detailed finite element investigation is indispensable. In most cases, polymeric products are investigated with multilinear material models over the entire stress-strain range. The definition of these material models requires material parameters, which must be obtained through time and cost demanding laboratory measurements. For some applications, such as quasi-static loading, it would be sufficient to use a linear elastic material model, the material constants of which can be obtained quickly. This article can help designers to apply the right material model as the linear elastic material model can be used until the 0.96% strain value in the case of the PA6 material with sufficient accuracy.**

**Abstract:** One of the most critical issues during polymer finite element simulations is the selection of the proper material models. The widely used and accepted multilinear material models require load case-specific material tests, which are time and cost demanding. Data for these characteristics must be acquired by standardized measurements. On the other hand, the parameters required to create a linear elastic material model in most cases are easy to obtain, and the establishment of the model is a shorter process. This research is aimed to provide information to engineers about the possibility of modeling the nonlinear elastic materials by using linear elastic material models and about the limits of such models. To create the most accurate material models, laboratory measurements were performed on polyamide (PA6) material, which is a widely used raw material in the industry. Test specimens were manufactured to obtain material constants according to the ISO 527-2 standard, and for validating the effectiveness of the applied material models, three different tensile specimens were created, which were tested under quasi-static loading in the elastic region. A comprehensive finite element investigation was performed, and the numerical results were then compared to laboratory measurements using the GOM Aramis digital image correlation (DIC) system. By comparing the optically measured strain data to the numerical results, it was determined that the nonlinear elastic materials can be modeled using linear elastic models in a well identifiable strain range with sufficient accuracy.

**Keywords:** polyamide 6; finite element analysis; digital image correlation; material model fitting

---

## 1. Introduction

Polyamide 6 (PA6) is a widely used material in many fields of the industry. It serves as the matrix of many composite materials, and it is one of the most significant materials in 3D printing. As this and many other polymeric materials are more and more frequently being used, engineers come across these materials in product design. Finite element simulations are now an essential part of any product's design phase. Different material models (elastic, plastic, viscoelastic, etc.) can simulate the material response of polymeric materials under

certain types of loadings. Since this research is aimed to help the design process, the plastic behavior was not investigated.

There are many accepted ways to model polymeric materials in the elastic region. The simplest approach is the use of a linear elastic material model, which is generally used to model the small strain behavior of the material. For the modeling of higher deformations in the elastic strain zone, the usage of nonlinear material models is advised (bilinear, multilinear and hyperelastic). The bilinear material model uses two lines, and the multilinear material model uses more than two lines to describe the stress-strain relation. Hyperelastic models (Mooney-Rivlin, Ogden, etc.) have been developed for elastomers. However, these models can also be used to model plastics. In the case of thermoplastics, such as PA6, hyperelastic models do not provide more information than simple linear elastic material models, see in [1].

The reason for the different modeling approaches is the nonlinear elastic relation between the stress and strain values of polymeric materials [2–5]. Generally, under small strains, the polymers behave linearly, but after a certain strain value, the linearity is lost, and the material starts to behave nonlinearly. The length of this section differs from polymer to polymer [6]. Another aspect, which must be considered is the viscoelastic behavior (linear or nonlinear) in polymeric materials, so the stress-strain behavior of the material is time dependent. In the case of linear viscoelastic materials, there is a linear connection between the stress and strain values at any given time, while nonlinear viscoelastic materials present a nonlinear connection. These materials are often characterized by their stress relaxation and creep behaviors [1,6–11]. PA6 is widely used as a structural element; thus, the time dependence of the material was neglected in our investigations because such elements are generally subjected to quasi-static loads. Alongside these properties, the mechanical properties of polymeric materials largely depend on the circumstances, such as the temperature, the testing rate, and the humidity [12–14].

The material that will be investigated is the PA6, a semicrystalline thermoplastic. Its stress-strain curves in the elastic region begin with a linear section, then transition into nonlinearity as the stresses increase [15,16]. The water content and the temperature also have a significant effect on the mechanical properties [14,16–18]. Testing rate dependence and creep are also present in the material [15,19]. To avoid measurement inconsistencies, all specimens have to be prepared from the same sheet of plastic, stored under the same conditions, and tested at the same rate.

The multilinear and viscoelastic material models are able to accurately model the nonlinear elastic behavior of the material, but the creation of them requires either load case-specific measurement data or, in the case of more complex loading, measurement data from different measurement methods [1,11,20–22]. The linear elastic material model, on the other hand, only requires Young's modulus or the shear modulus and the Poisson's ratio of the material [1]. These data are broadly available, and the creation of the material model is simpler, but the effectiveness of these models is varied [6].

The different finite element simulations can be validated by laboratory experiments using full-field optical measuring systems, which work based on the digital image correlation (DIC) method. These systems create pictures during the measurement and compare the changes of the required stochastic pattern to calculate various measurements, such as deformation and strain. There are two-dimensional and three-dimensional DIC measuring systems. This equipment is widely used in material and structure testing, because of the ease of setting up a measurement and because of its ability to measure along the whole surface instead of only between two points [23,24]. In material testing, it is widely used to evaluate the different local strains and deformations on the surface of the specimen [25–27]. The results from these experiments can be directly compared to the results of accurately created finite element simulations [5,11,28–31]. By this comparison, the different material models can also be validated, and the most suitable ones can be selected [32,33].

The comparison between the simulations and the optical measurements not only provides information about the most suitable material model, but when tested at both

lower and higher, but still elastic strains, it also outlines the limits and capabilities of each material model. This work is aimed to find out the effectiveness of the linear elastic material models when modeling PA6 by comparing them to a nonlinear elastic material model and validating all results using advanced full-field measurement systems. This material was chosen because its stress-strain curves have a significant amount of linearity and nonlinearity as well, and it is widely used in many fields of the industry. During the measurements, only the elastic properties were tested, the measurement method was set up in a way to avoid any other factors, such as testing rate dependence, water content dependence, and viscous properties, to simulate the effect of a quasi-static load. By finding out the limits of the linear elastic material models when modeling nonlinear elastic materials under quasi-static loading, the possibilities of these material models' usage in the industry can be determined, thus shortening the time and effort required to set up a finite element simulation for such materials. During the product development phase, only the elastic region of the material is in focus because any kind of failure has to be avoided.

## 2. Materials and Methods

### 2.1. Specimens for Mechanical Testing

To compare the material models as accurately as possible, a sheet of PA6 was purchased to be used for the tensile test by the ISO 527-2 standard and for the validation of the material models. For the standardized tensile testing, five specimens were cut out from the 2 mm thick sheet using the INSTRON CEAST punching machine according to the ISO 527-2 Type 1A geometry, which has a width of 10 mm at the tested length. The specimens were also marked lengthwise by the gauge length of 75 mm, recommended by the ISO 527-2 standard [34]. These points were used by the INSTRON AVE2 video extensometer to measure the axial strain values of the specimen. Being a biaxial measurement system, the AVE2 can also measure the crosswise strain values; thus, two additional points were added. The second axis was used to calculate the Poisson's ratio of the material. A marked specimen can be seen in Figure 1.

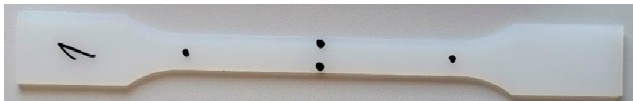

**Figure 1.** Prepared ISO 527-2 Type 1A PA6 tensile specimen.

To validate the effectiveness of the material models, three different tensile specimen geometries were created to test each material model under different stress states. The geometries were created to induce different stress states inside the specimen [35]. The specimen geometries can be seen in Figure 2.

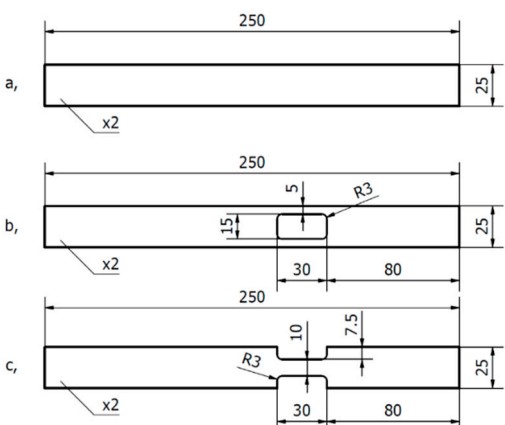

**Figure 2.** Specimen geometries for the material model validation: (**a**) simple plate, (**b**) middle-weakened plate, (**c**) side-weakened plate (all sizes are given in mm).

Three of each specimen geometry was manufactured from the same sheet of PA6. The manufactured specimens can be seen in Figure 3.

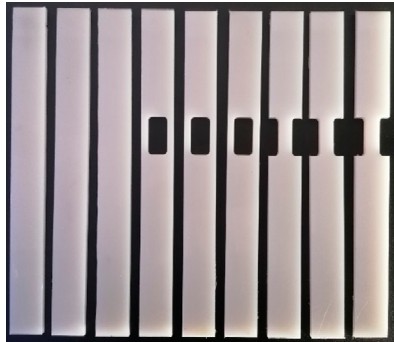

**Figure 3.** Manufactured specimens for the material model validation.

### 2.2. Mechanical Testing with Video Extensometer and Optical Measuring Device

The material testing was performed using the INSTRON 68TM-10 universal testing machine and the INSTRON AVE2 video extensometer. The measurement is controlled and evaluated by the Bluehill software, which already has standardized measurement methods defined. The ISO 527-2 method was used for this measurement. For the gripping of the ISO 527-2 Type 1A tensile specimen, wedge action tensile grips were used. The testing speed was chosen as 50 mm/min from the ISO 527-2 standard, and the tests lasted until the complete failure of the specimen [34]. The same testing rate was used for all measurements to avoid inconsistencies. The complete testing setup can be seen in Figure 4. After breaking all the specimens, the results were evaluated based on the ISO 527-2 standard and based on what is required for the finite element simulations [34].

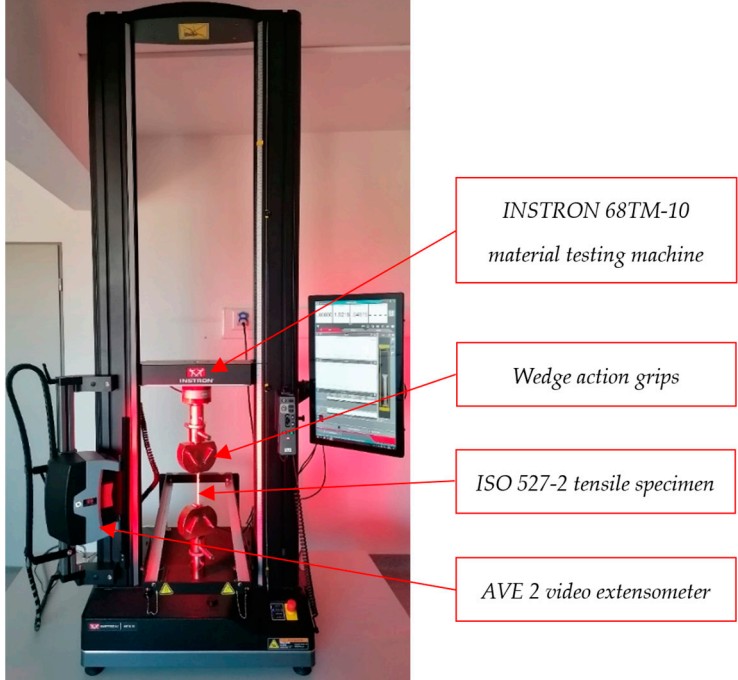

**Figure 4.** Testing setup for tensile testing.

For the material model validation, the specimens seen in Figure 3 were used. In this case, the elastic behavior of the material was tested, so the load was maximized at a value that is about 25% less than the measured yield strength. The maximum force was then calculated, and another safety factor was added because, along the radiuses of the more

complex geometries, stress concentration will occur, which could result in the material exceeding the maximum load. Thus, the maximum load of the specimen, in the end, was defined as 400 N; the moment of reaching this force value marks the end of the test.

Laboratory measurements for the material model validation were performed using the same material testing machine and the GOM Aramis AdjusTable 12M full-field optical measurement system. The Aramis system uses two cameras to create high-definition images in 3D. It is required to paint a stochastic pattern on the specimen's investigated surface. Two different paints were used, the first layer was a removable matte white coat, and the second layer was just simple black acrylic paint. The movement of the speckles compared to the reference image was evaluated as various measurements with the GOM Correlate Professional software [36].

The measurement method was set up in a way that after the gripping of the specimen, one picture was created by the camera, then the material testing machine applied the 400 N load, with a 50 mm/min testing rate. At the point of reaching the maximum load, a second picture was created by the Aramis. Figure 5 shows the measurement setup.

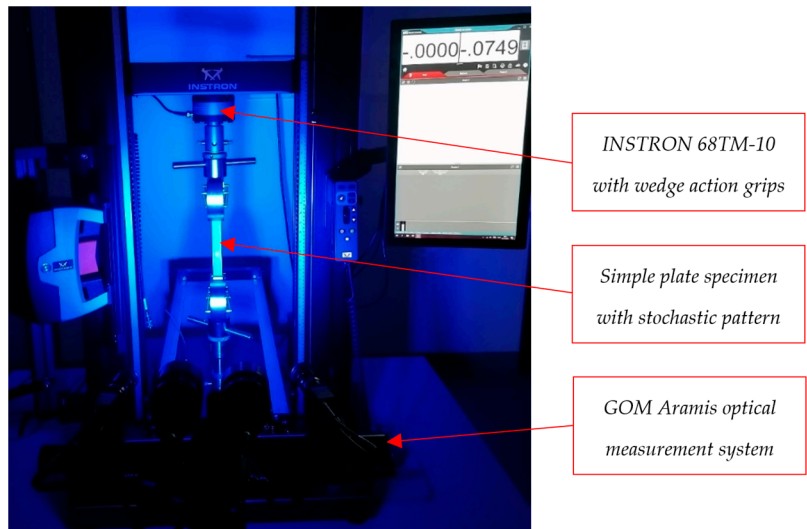

INSTRON 68TM-10
with wedge action grips

Simple plate specimen
with stochastic pattern

GOM Aramis optical
measurement system

**Figure 5.** Measurement set up for the validation of the material models using the GOM Aramis.

### 2.3. Finite Element Analysis

To perform the finite element simulations, the Simcenter Femap software was selected. Its solver offers a simple linear elastic material model and a multilinear material model as well. This model is defined by the input of measured stress-strain data points [37].

The specimen geometries were created in the finite element software as two-dimensional models. The gripping length of 50 mm was removed from both ends of all the models, so the loads and constraints can be added to the curve. The constraints and loads of the specimen were defined to mimic the ones induced by the testing machine. After creating the models as accurately as possible, the material models were also defined. The models were meshed using linear four-node quadrilateral plane stress elements, and then a mesh convergence analysis was performed to investigate the mesh independence of the calculated results. The analysis has shown that the 0.25 mm average element size is optimal for this simulation. Using these element attributes, the simple plate specimen was meshed by 60,000 elements, the middle-weakened plate by 52,929 elements, and the side-weakened plate by 52,900 elements. Finally, the simulations were run with the selected element size.

The results from all the finite element simulations and the laboratory experiments have been post-processed and compared. To ensure the most accurate comparison between the two methods, a smaller (Zone 1) and a higher strain zone (Zone 2) was defined on the specimen, see Figure 6. These zones were used to calculate the average strain values. On these zones, relative errors have been calculated based on the experimental results.

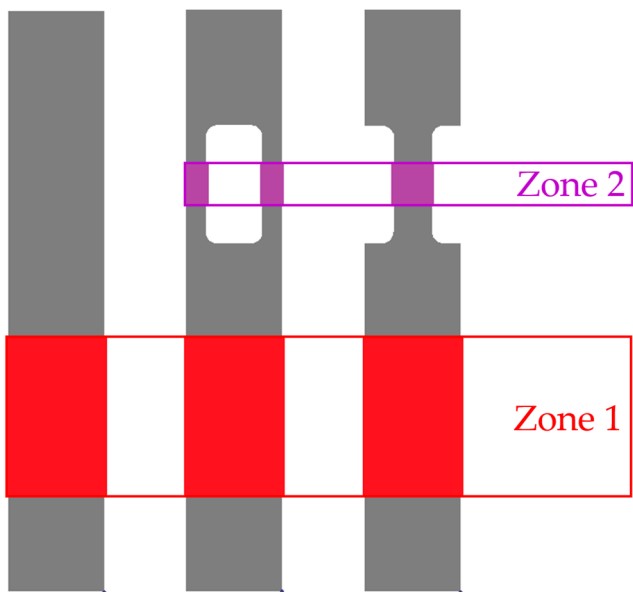

**Figure 6.** Zones defined for evaluation of the average values.

## 3. Results and Discussion

### 3.1. Results of the ISO 527-2 Tensile Tests

The measured stress-strain curves of the tensile test of the ISO 527-2 Type 1A specimens can be seen in Figure 7.

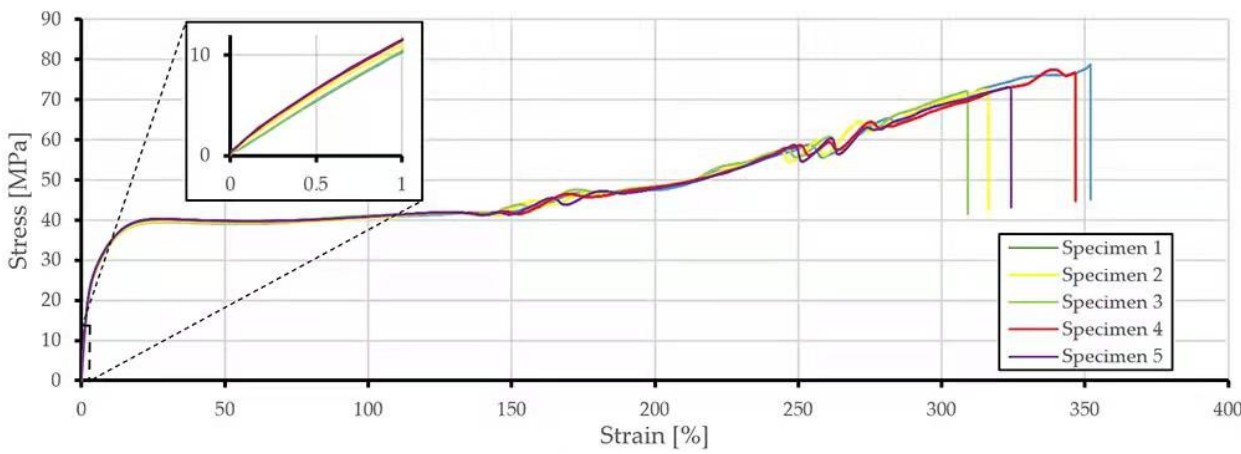

**Figure 7.** Stress-strain curves of the five tensile specimens until failure.

From the measured characteristics, Young's modulus and the yield stress were evaluated. First, the Young's modulus was evaluated at the strain interval of 0.05–0.25% recommended by the ISO 527-2 standard [34]. However, the calculated results' maximum deviation from the median value was way more than 5%. To check if it was a measurement error or the material had inconsistencies in that small strain region, the Young's modulus was checked between the 0.25% and 0.45% strain values. This strain interval provided significantly better results. The Young's modulus values of the five specimens in the two different strain intervals can be seen in Table 1.

**Table 1.** Young's modulus values of the five specimens in the two strain intervals.

| Specimens | Young's Modulus Values at 0.05–0.25% Strains (MPa) | Young's Modulus Values at 0.25–0.45% Strains (MPa) |
| --- | --- | --- |
| Specimen 1 | 1079.000 | 1077.700 |
| Specimen 2 | 1261.000 | 1099.900 |
| Specimen 3 | 1079.200 | 1100.800 |
| Specimen 4 | 1330.300 | 1122.100 |
| Specimen 5 | 1362.000 | 1134.800 |
| Average | 1222.300 | 1107.060 |
| Std. deviation | 135.730 | 22.069 |
| Median | 1261.000 | 1100.800 |
| Max. difference | 14.430% | 3.090% |

From Table 1, it can be seen that the smaller interval provided significantly higher standard deviation and maximum difference values. Since the 0.25–0.45% strain interval provided results with less than 5% maximum difference, those values were used in the simulations.

After evaluating the average Young's modulus, the average yield stress and the average Poisson's ratio were also evaluated for the simulations. These values can be seen in Table 2.

**Table 2.** Measured average material constants required for the simulations.

| Material Constants | Average Value |
| --- | --- |
| Young's modulus (MPa) | 1107.060 |
| Yield stress (MPa) | 39.981 |
| Poisson's ratio (-) | 0.450 |

After evaluating the constants required for the simulations, the elastic behavior of the material was inspected. By fitting a line based on the evaluated Young's modulus to each specimen and calculating the relative difference between the measured curve and the fitted line, we were able to determine when the linear elastic region ends. The maximum relative error was defined as 5%. The first strain value where the relative error becomes larger than the defined maximum has been chosen as the end of the linear elastic section. These values can be seen in Table 3.

**Table 3.** The end of the linear elastic section for each specimen based on a fitted line.

| Specimens | End of the Linear Elastic Region (%) |
| --- | --- |
| Specimen 1 | 0.976 |
| Specimen 2 | 0.956 |
| Specimen 3 | 0.926 |
| Specimen 4 | 0.995 |
| Specimen 5 | 0.961 |
| Average | 0.963 |
| Std. deviation | 0.026 |
| Median | 0.961 |
| Max. difference | 3.69% |

From Table 3, it can be seen that the end of the linear elastic region in the material is at a strain value of 0.96%. This strain value is the expected limit of the linear elastic material model's accuracy.

### 3.2. Material Models Based on the Measured Results

Three material models were used for the simulations. The first and most simple one is the linear elastic material model based on the Young's modulus calculated by the ISO 527-2 tensile test [34]. The input values for this material model were the Young's modulus and Poisson's ratio values from Table 2.

The second material model was another linear elastic material model. In this case, the Young's modulus was calculated until the stress limit of the material model validated measurement. The limit was set to be 30 MPa to ensure that the specimen remains in the elastic region, as explained in Section 2.2. The values were calculated using the following equation:

$$E = \frac{\sigma_{30} - \sigma_0}{\varepsilon_{30} - \varepsilon_0}, \tag{1}$$

where $E$ is the Young's modulus, $\sigma_{30}$ is the closest stress value to 30 MPa, $\sigma_0$ is the lowest stress value of the curve, $\varepsilon_{30}$ is the strain at the 30 MPa stress value and the $\varepsilon_0$ is the lowest strain value of the curve. The results of the calculations for each specimen can be seen in Table 4.

**Table 4.** Young's modulus values until the limit stress value.

| Specimens | Young's Modulus until the Limit Stress (MPa) |
|---|---|
| Specimen 1 | 447.757 |
| Specimen 2 | 450.420 |
| Specimen 3 | 461.211 |
| Specimen 4 | 477.030 |
| Specimen 5 | 478.626 |
| Average | 463.009 |
| Std. deviation | 14.446 |
| Median | 461.211 |
| Max. difference | 3.78% |

Because of the small standard deviation and maximum difference, the average value can be used for further calculations. The material constants for the second material model can be seen in Table 5.

**Table 5.** The material constants required to define the linear elastic material model with the Young's modulus calculated until the limit stress.

| Material Constants | Value |
|---|---|
| Young's modulus (MPa) | 463.009 |
| Poisson's ratio (-) | 0.450 |

The third material model was the nonlinear elastic material model available in the NX Nastran solver. This model is a multilinear material model and is defined by creating a stress-strain function from the measured stress-strain values in the software. To define this material model, the median specimen was selected based on the measured values. The stress-strain data of this specimen were selected until the yield stress and then reduced into 30 data points to ensure a quicker and easier calculation for the software. These data points were then placed into a function in Femap, which was added to the material model along with the Young's modulus and Poisson's ratio values, see in Table 2. The defined nonlinear curve can be seen in Figure 8, along with the other stress-strain relations of the other two used material models.

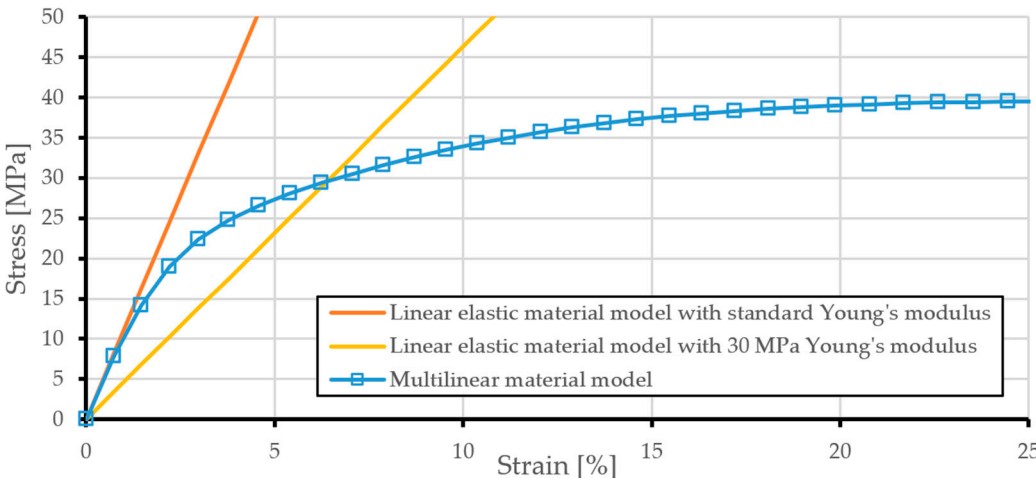

**Figure 8.** The stress-strain curves of the three material models.

### 3.3. Results of the Finite Element Simulations

The finite element simulations were run using the 0.25 mm element size and the NX Nastran solver. In the case of the two linear elastic material models, a static simulation was used, and in the case of the multilinear material model, a nonlinear analysis was used with 100 time steps.

The simulation results were evaluated, and the principal strains and deformations can be seen in Figures 9 and 10, respectively.

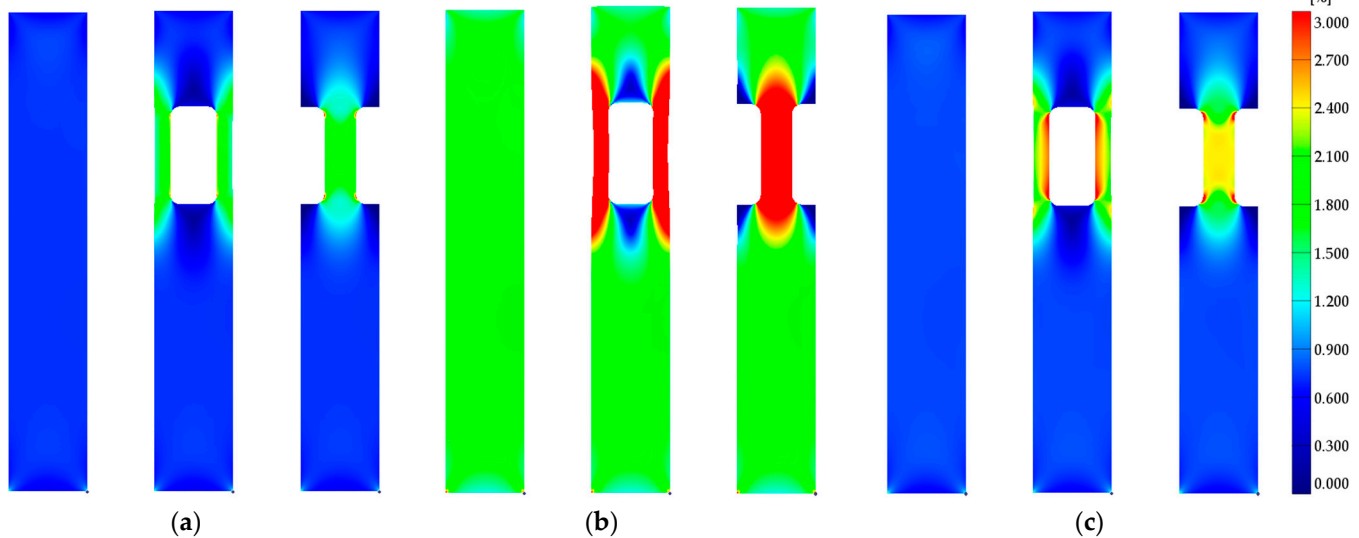

**Figure 9.** Strain results of the three material models. (**a**) The linear elastic material model with standard Young's modulus. (**b**) The linear elastic material model with Young's modulus until limit stress. (**c**) The multilinear material model.

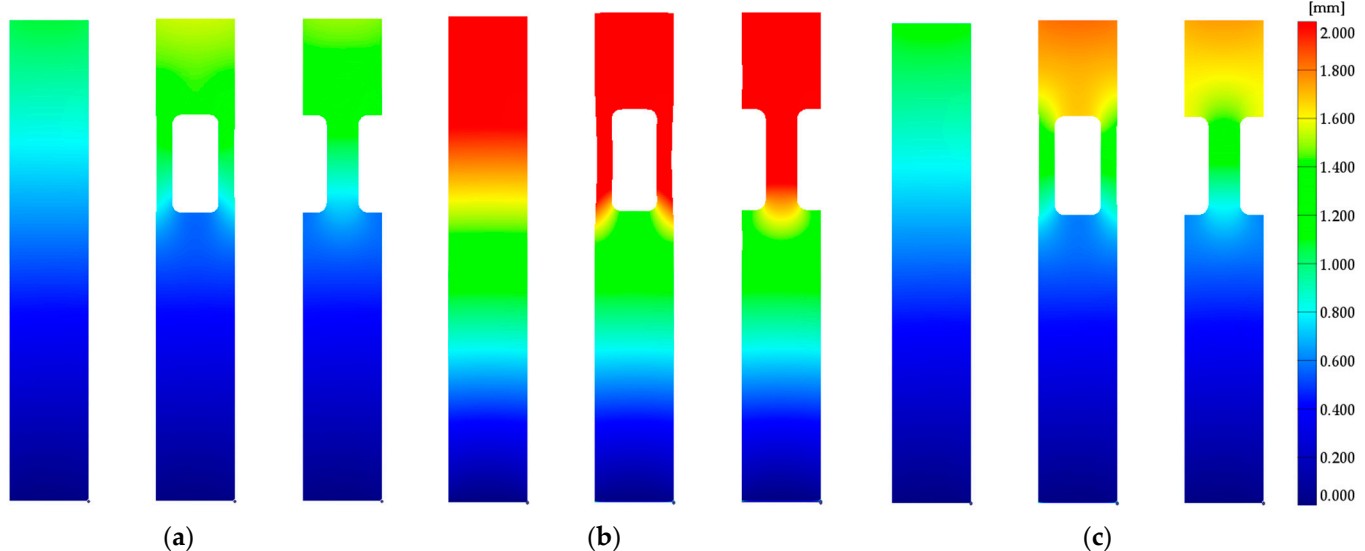

**Figure 10.** Deformation results of the three material models. (**a**) The linear elastic material model with standard Young's modulus. (**b**) The linear elastic material model with Young's modulus until limit stress. (**c**) The multilinear material model.

The simulated data were further evaluated during the comparison with the optical measurement results.

### 3.4. Results of the Optical Measurements

The optical measurement was performed on all nine specimens. Based on the average strain values of Zone 1 (defined in Figure 6), the median values of all three specimen geometries were chosen for further evaluation. The average values can be seen in Table 6.

**Table 6.** Average strain values of the 9 specimens in Zone 1.

| Specimens | Simple Plate | Middle-Weakened Plate | Side-Weakened Plate |
| --- | --- | --- | --- |
| Specimen 1 | 0.744 | 0.727 | 0.794 |
| Specimen 2 | 0.745 | 0.745 | 0.775 |
| Specimen 3 | 0.782 | 0.733 | 0.774 |
| Average | 0.757 | 0.735 | 0.781 |
| Std. deviation | 0.0217 | 0.00917 | 0.0113 |
| Median | 0.745 | 0.733 | 0.775 |
| Max. difference | 4.97% | 1.64% | 2.45% |

Based on these results, specimen 2 was selected from the simple and side-weakened plates and specimen 3 from the middle-weakened plate. These three specimens were used in the comparison. The strain and deformation results of these three geometries can be seen in Figures 11 and 12.

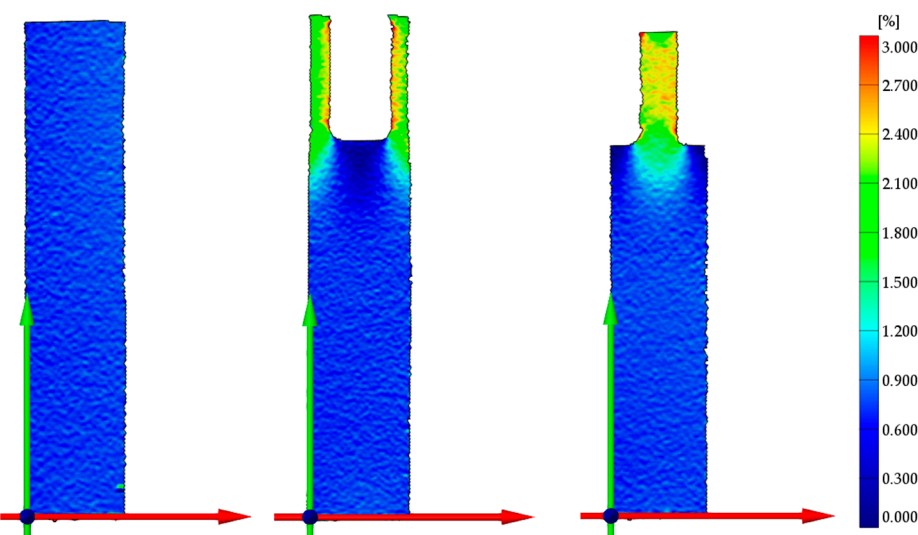

**Figure 11.** Strain results of the optical measurements.

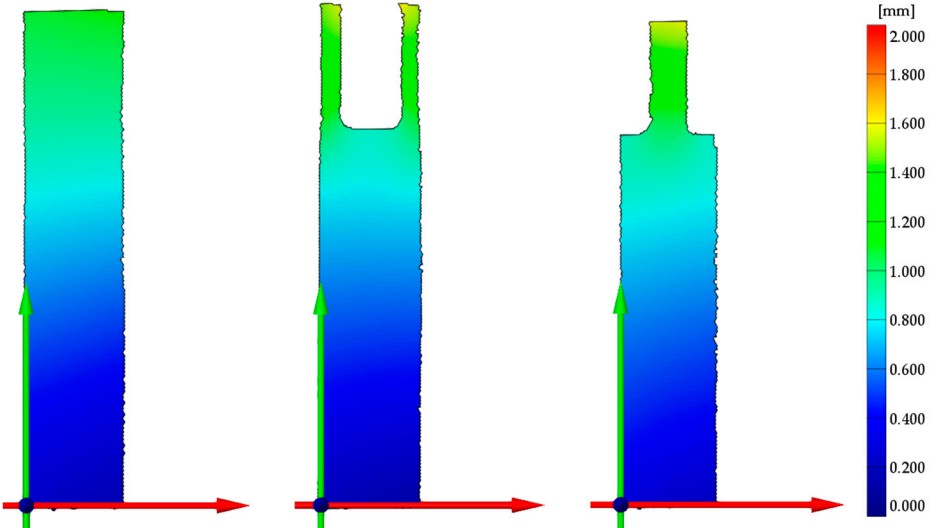

**Figure 12.** Deformation results of the optical measurements.

### 3.5. Comparison of the Numerical and Optical Results

The results of the two methods were compared. Relative errors were calculated to evaluate the accuracy of the models. The optical measurement results were used for the base of the relative error calculations.

#### 3.5.1. Comparison of the Average Strain Values in Zone 1

For this comparison, a simple plate was used. The evaluated zone was selected as Zone 1, see in Figure 6. This zone is 40 mm long and starts 25 mm away from the fixed grip. The evaluated values were organized into a table alongside the relative errors, which are given in Table 7. The relative errors were calculated using the following equation:

$$e_r = \frac{v_{simulated} - v_{measured}}{v_{measured}} \cdot 100 \ [\%], \tag{2}$$

where $e_r$ is the relative error, $v_{simulated}$ is either the strain or displacement value evaluated from the simulations, and $v_{measured}$ is either the strain or the displacement value evaluated from the optical measurements. This equation was used for the calculation of all relative error values.

**Table 7.** Average strain values of the simple plate and the relative errors based on the optical measurement.

| Methods | Average Strain (%) | Relative Errors |
|---|---|---|
| Optical measurement | 0.745 | - |
| The linear elastic material model. with standard Young's modulus | 0.721 | −3.22% |
| The linear elastic material model with Young's modulus until limit stress | 1.728 | 131.96% |
| Multilinear | 0.766 | 2.82% |

It can be seen from these results that both the linear elastic material model with standard Young's modulus and the multilinear material model provided results with less than 5% relative errors. These results can be considered as accurate. The images of the four methods can be seen in Figure 13.

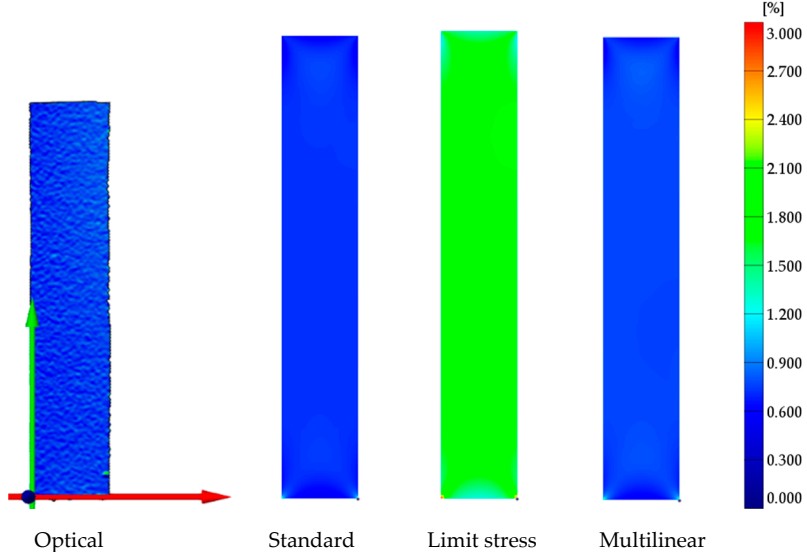

Optical Standard Limit stress Multilinear

**Figure 13.** The four strain images of the simple plate.

3.5.2. Comparison of the Average Strain Values in Zone 2

For this comparison, the middle-and side-weakened plates were used. In this case, Zone 2 was evaluated, see in Figure 6. These zones are 10 mm long and are positioned in the middle of the length of the smaller cross-sections. The results of each geometry were organized into tables, which can be seen in Table 8.

**Table 8.** Average strain values of the middle- and side-weakened plates at Zone 2 and the relative errors based on the optical measurement.

| Methods | Middle-Weakened Plate | | Side-Weakened Plate | |
|---|---|---|---|---|
| | Average Strain | Relative Error | Average Strain | Relative Error |
| | (%) | (%) | (%) | (%) |
| Optical | 2.335 | - | 2.396 | - |
| Standard | 1.802 | −22.82 | 1.825 | −23.84 |
| Limit stress | 4.320 | 84.99 | 4.320 | 80.31 |
| Multilinear | 2.441 | 4.52 | 2.426 | 1.24 |

The results show that in the case of higher strain values, only the multilinear material model was able to provide accurate results. As an example, the strain images of the side-weakened plate can be seen in Figure 14.

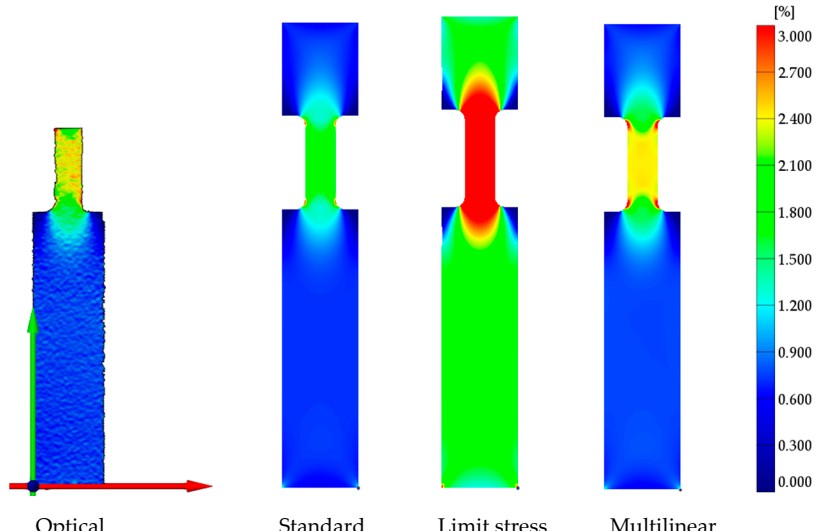

**Figure 14.** The four strain images of the side-weakened plate.

### 3.5.3. Comparison of the Deformation Values

The deformation values were evaluated on all three specimen geometries at the 120 mm length, and the relative errors were also determined, see in Table 9.

**Table 9.** Deformation values of the three plates at the 120 mm length and the relative errors based on the optical measurement.

| Methods | Simple Plate | | Middle-Weakened Plate | | Side-Weakened Plate | |
|---|---|---|---|---|---|---|
| | Deformation | Relative Error | Deformation | Relative Error | Deformation | Relative Error |
| | (mm) | (%) | (mm) | (%) | (mm) | (%) |
| Optical | 0.961 | - | 1.631 | - | 1.521 | - |
| Standard | 0.861 | −10.37 | 1.410 | −13.56 | 1.349 | −11.32 |
| Limit stress | 2.065 | 114.82 | 3.379 | 107.17 | 3.233 | 112.54 |
| Multilinear | 0.915 | −4.77 | 1.698 | 4.11 | 1.591 | 4.62 |

Based on the results in all cases, only the multilinear material model provided accurate results. As an example, the deformation images of the middle-weakened plate can be seen in Figure 15.

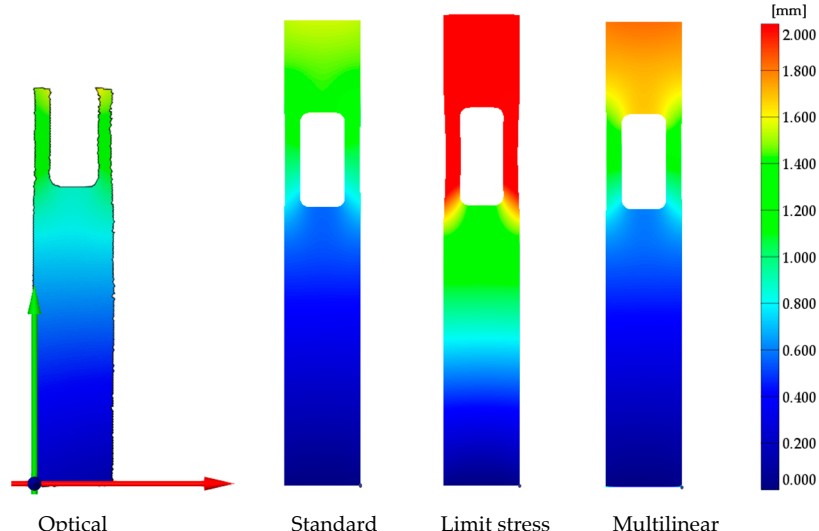

**Figure 15.** The four deformation images of the middle-weakened plate.

## 4. Conclusions

This article investigated the capabilities of linear elastic material models when modeling a nonlinear elastic material under quasi-static loading. First, a tensile test was performed by the ISO 527-2 standard to ensure that the defined material models were as accurate as possible. During the evaluation of the standard measurement results, the Young's modulus calculated in the ISO 527-2 standard's recommended 0.05–0.25% strain interval showed higher than 10% error values. Thus, the Young's modulus for the material was determined at the 0.25–0.45% strain interval with a 3% error. Based on the material testing results, a linear elastic material model defined by the standard Young's modulus, a linear elastic material model with Young's modulus calculated until the limit stress, and a multilinear material model was chosen for the finite element simulations. The models were then simulated using three different specimen geometries. After the simulations, the laboratory measurements were performed using the GOM Aramis full-field optical measurement system. The results of the comparison showed that the specimen geometries and the method of comparing the optical measurements are accurate and usable for the validation of material models.

The comparison between the three material models and the optical measurements showed that the linear elastic material models with the standard Young's modulus can be used for the modeling of the small strain behavior of nonlinearly elastic materials. In the case of the PA6 material, the limit of modeling is 0.96% strain. It is interesting to note that during comparison, this material model showed the same 3% error that was seen when calculating the standard Young's modulus of the material, meaning that the 3% error could exist in the production of the sheet material and not in the modeling itself. When evaluating the deformations, on the other hand, the linear elastic material models never provided accurate results. This could be due to the fact that near the grips, increased strain values occur, which exceed the modeling limit of 0.96% strain. This results in increased local deformations, which the material model is not able to account for.

Overall, the linear elastic material model with standard Young's modulus could be used in the industry to model any polymeric materials' small strain behavior when the product is under quasi-static loading, and the goal is to find the average strain or stress values of a part. In these cases, the usage of this material model can shorten the design process because setting up such a model is a shorter process than creating a multilinear material model, and the data required for this model is vastly available. In contrast, the stress-strain curves required to create the multilinear material models must be obtained via standardized material testing, which is a costly and time-consuming process. Although, to evaluate deformations, the multilinear material models are still the more accurate choice.

**Author Contributions:** Conceptualization, D.H. and T.M.; methodology, M.F. and D.H.; software, M.F.; validation, D.H.; formal analysis, M.F.; investigation, D.H. and M.F.; resources, T.M.; data curation, M.F.; writing—original draft preparation, M.F.; writing—review and editing, D.H. and T.M.; visualization, M.F.; supervision, T.M.; project administration, T.M.; funding acquisition, T.M. All authors have read and agreed to the published version of the manuscript.

**Funding:** The research was supported by the Thematic Excellence Programme (TKP2020-NKA-04) of the Ministry for Innovation and Technology in Hungary, within the framework of the (Automotive Industry) thematic program of the University of Debrecen.

**Institutional Review Board Statement:** Not applicable.

**Informed Consent Statement:** Not applicable.

**Conflicts of Interest:** The authors declare no conflict of interest associated with this publication.

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
