# Peer review of "Optical Investigation of the Limits of Modeling the Nonlinear Elastic Behavior of PA6 Using Linear Elastic Material Models"

_applsci, doi:10.3390/app12031057_

Round 1

Reviewer 1 Report

In the attachment

Reviewer 2 Report

Minor Comments:

  1. Figure 7 is good. Please, zoom the portion of the strain range from 0 to 1%, because this will be clearer for the reader and show your data collection range.
  2. Fig. 2, what is the unit?
  3. Line 166-169: How many elements did you use? How many nodes per element? What are the final dimensions of the element you used?
  4. Fig 4, Table 1, Table 3, and so on, you showed the result of five specimens, which is good. Can you tell the reader somewhere in the manuscript what type of specimen you used, i.e., is it type a, b or c from Fig6?

Round 2

Reviewer 1 Report

Manuscript ID: applsci – 1537356

Title: Optical investigation of the limits of modeling the nonlinear elastic behavior of PA6 using linear elastic material models

The paper can be accepted for publication after a minor revision.

P2, line 57: In the case of thermoplastics, such as PA6, hyperelastic models do not provide more information than simple linear elastic material models, see in [1].

P2, line 72: PA6 is widely used as a structural element, thus the time dependence of the material was neglected in our investigations because such elements are generally subjected to quasi-static loads.

P7, line 196: The analysis has shown

P14, line 316: are given in Table 7

P15, Table 8: Average strain (%) or Deformation (mm)?

P15, line 337: multilinear
